# ReSimAD: Zero-Shot 3D Domain Transfer for Autonomous Driving with Source Reconstruction and Target Simulation

**Bo Zhang**[1,*]**, Xinyu Cai**[1,*✉]**, Jiakang Yuan**[3,1]**, Donglin Yang**[4]**, Jianfei Guo**[1]**, Xiangchao Yan**[1]**,
Renqiu Xia**[2,1]**, Botian Shi**[1,‡]**, Min Dou**[1]**, Tao Chen**[3]**, Si Liu**[4]**, Junchi Yan**[2,1✉]**, Yu Qiao**[1]

[1] Shanghai Artificial Intelligence Laboratory, [2] Shanghai Jiao Tong University
[3] Fudan University, [4] Beihang University
* Equal Contribution, ✉ Corresponding Authors, ‡ Project Leader

## Abstract

Domain shifts such as sensor type changes and geographical situation variations are prevalent in Autonomous Driving (AD), which poses a challenge since AD model relying on the previous domain knowledge can be hardly directly deployed to a new domain without additional costs. In this paper, we provide a new perspective and approach of alleviating the domain shifts, by proposing a Reconstruction-Simulation-Perception (ReSimAD) scheme. Specifically, the implicit reconstruction process is based on the knowledge from the previous old domain, aiming to convert the domain-related knowledge into domain-invariant representations, *e.g.*, 3D scene-level meshes. Besides, the point clouds simulation process of multiple new domains is conditioned on the above reconstructed 3D meshes, where the target-domain-like simulation samples can be obtained, thus reducing the cost of collecting and annotating new-domain data for the subsequent perception process. For experiments, we consider different cross-domain situations such as Waymo-to-KITTI, Waymo-to-nuScenes, *etc*, to verify the **zero-shot** target-domain perception using ReSimAD. Results demonstrate that our method is beneficial to boost the domain generalization ability, even promising for 3D pre-training. Code and simulated points are available at: https://github.com/PJLab-ADG/3DTrans.

## 1 Introduction

Autonomous Driving (AD) aims to perform the perception of the ego-car's surroundings and further make decisions without human involvement (Sun et al., 2020; Li et al., 2023). In recent years, an increasing number of self-driving vehicles at the L2 level are gradually entering our lives, becoming an indispensable part of traffic elements (Mozaffari et al., 2020). Although it has achieved promising progress, there are still many issues that need to be addressed, such as continuous data collection and annotation efforts, before accomplishing a robust autonomous driving system.

Considering a more demanding yet practical cross-domain scenario: As a manufacturer of AD system, the self-driving vehicle product is required to be updated to the next version for business purposes. However, this product update process may encounter the following challenges caused by domain shifts. The base model of the current-version self-driving vehicle is well-trained on the massive labeled source domain collected on the previous old domain knowledge (such as previous sensor technologies or data acquisition cities), but it needs to achieve a good performance on different domains such as next-generation sensor technology or unseen streets. Unfortunately, previous works (Yang et al., 2022; Wei et al., 2022; Yuan et al., 2023b; Huang et al., 2023; Xu et al., 2021a; Geiger et al., 2012; Wang et al., 2020; Dong et al., 2023) pointed out that the existing AD models, including 3D detection (Shi et al., 2021; Deng et al., 2021), segmentation (Hou et al., 2022; Graham et al., 2018), motion (Shi et al., 2022), and planning models (Teng et al., 2023), typically face serious performance drop on the target domain with such cross-domain shifts.

One commonly used solution to transfer the base model from its original domain to a new target domain is the supervised fine-tuning on the new target domain (Yin et al., 2021; Yuan et al., 2023a),

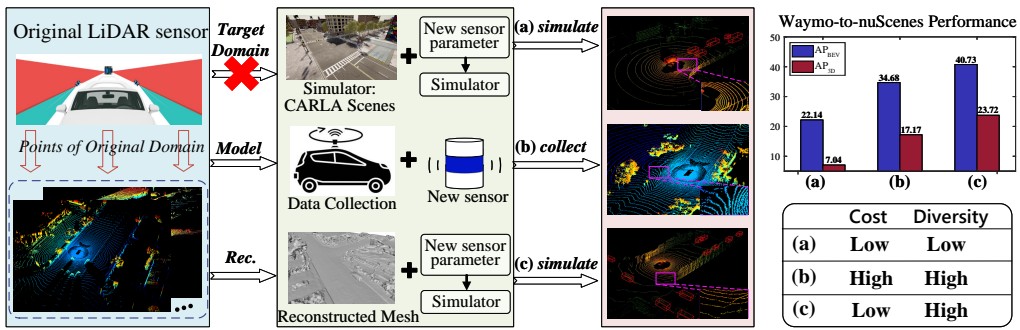

Figure 1: Different paradigms for cross-domain Autonomous Driving (AD), where Rec. denotes the employed reconstruction scheme. (a) By directly simulating target-domain data from CARLA (Dosovitskiy et al., 2017), it expands the number of training samples from the target domain. But the diversity of the simulated data directly from CARLA is low. (b) Other works (Yang et al., 2021; 2022; Yuan et al., 2023b; Wei et al., 2022) often employ Unsupervised Domain Adaptation (UDA) to enable learning from target-domain data distribution. But this process needs to collect massive target-domain samples from the real world, which is expensive. (c) In the proposed ReSimAD paradigm, the well-labeled data from the old domain is utilized to reconstruct the 3D scene, and then, the target-domain-like data is simulated from the reconstructed scene.

which is expensive due to that it needs to pay a high price for **data acquisition** and **human annotation** of the target domain. Besides, the long development cycle of data acquisition and human annotation on a new domain (*e.g.*, next-generation sensor for self-driving vehicles) could give rise to the delay in the product delivery of the next-generation self-driving vehicle.

Recently, researchers have tried to bridge the domain gaps, by either rendering autonomous driving scenes directly from the simulation engine (Dosovitskiy et al., 2017) to reduce the **data acquisition** cost, or leveraging Unsupervised Domain Adaptation (UDA) technique (Yang et al., 2021; Wei et al., 2022; Yuan et al., 2023b; Zhang et al., 2023; Xu et al., 2021a; Yang et al., 2022) to reduce **human annotation** cost, which is illustrated in Fig. 1.

Different from the above-mentioned research works, we consider the particular issue faced by the AD manufacturer: How to achieve a zero-shot source-to-target model transfer. **Note that** the *zero-shot* means that for the target domain, there is almost no additional data acquisition and labeling cost. To achieve this goal, we propose ReSimAD – a unified perception-simulation-perception pipeline. Given the access to the annotations from the source domain, we propose to leverage the source-domain annotated data to reconstruct the 3D real scene, producing the 3D mesh to decouple the domain characteristics. Further, the 3D mesh can be fed into the simulation engine to re-inject some target domain-related characteristics such as LiDAR sensor parameter setting, according to minor prior information that we have obtained about the target domain.

Empirically, we obtain the target-domain-like points by the reconstruction-simulation scheme, showing the effectiveness of the simulated points on the real target-domain scenario. We conduct experiments under Waymo-to-KITTI, Waymo-to-nuScenes, Waymo-to-ONCE settings, and the results show excellent zero-shot ability when the perception model faces the domain shift, even surpassing some UDA baselines that have access to extensive real-world target domain data.

**Contribution.** **(1)** We provide the autonomous driving community with the knowledge that, the scheme of source-domain reconstruction followed by target-domain simulation essentially improves the robustness for an unseen target domain. **(2)** We propose ReSimAD, a unified reconstruction-simulation-perception paradigm that addresses the domain shift issue, where the reconstructed 3D meshes decouple the domain characteristics, bridging the well-labeled old domain and an unseen new domain. **(3)** By extensive experiments on different datasets with distinct domain shifts, ReSimAD achieves high 3D detection accuracy under zero-shot target-domain setting, even outperforming the unsupervised domain adaptation method that has to access massive real target domain data.

## 2   RELATED WORKS

**3D Object Detection under Different Domains.** The domain variations make the model trained on the fully-labeled source domain hard to achieve a satisfactory detection accuracy on the target domain with domain differences. Notably, the idea of Domain Adaptation (DA) commu-

Figure 2: Visualization results between (a) real domain and (b) simulated domain. The domain simulated by ReSimAD is close to the real domain, such as slope on the road.

nity (Long et al., 2015; Fu et al., 2021) has been well developed, especially for the 3D cross-domain works (Yuan et al., 2023b; Zhang et al., 2023; Yang et al., 2021; Wei et al., 2022; Xu et al., 2021a) that handle the domain shifts on the Autonomous Driving (AD) scene. For example, ST3D (Yang et al., 2021) and LiDAR-Distillation (Wei et al., 2022) design a self-training pseudo-labeling pipeline and beam distillation method, respectively, assuming that the real data from the target domain are available for the training process. Besides, Bi3D (Yuan et al., 2023b) proposes to actively sample both the source and target samples to reduce the domain gap. Recently, Uni3D (Zhang et al., 2023) extends the study of cross-domain adaptation, investigating the possibility of joint training on multiple merged point datasets. **Different from previous DA study works**, we address the domain gaps from another perspective: 1) domain-decoupling based on the old-domain data, and 2) re-sampling new-domain-like data based on a simulator in a cost-free manner.

**Scene Reconstruction and Simulation.** Environment simulation (Dosovitskiy et al., 2017) is often used in AD scenes to reduce the costs of collecting expensive training data. Early works (Cai et al., 2023; Dosovitskiy et al., 2017; Gschwandtner et al., 2011) mainly focus on leveraging the virtual sensor simulation such as CARLA (Dosovitskiy et al., 2017) or Blensor (Gschwandtner et al., 2011). For example, by utilizing a virtual sensor that can be configured to simulate real-world devices, additional synthetic data can be generated for model training (Wang et al., 2019). Recently, researchers have started to use real data to strengthen the realism of the simulation process under the complex world. For example, LiDARsim (Manivasagam et al., 2020) builds an asset Library of 3D static maps and 3D dynamic objects, by means of driving around several cities in the real world. UniSim (Yang et al., 2023) proposes a closed-loop multi-sensor simulation scheme, which reconstructs the dynamic objects and static background, further producing different observations of new viewpoint. **Different from these scene reconstruction and simulation methods**, our ReSimAD fully investigates how to boost the zero-shot target-domain perception performance using the reconstruction-simulation pipeline, where we separate the reconstruction and simulation process, allowing the model to simulate wider data distribution variations.

## 3 THE RECONSTRUCTION-SIMULATION DATASET

To compare with recent research works (Yang et al., 2021; Yuan et al., 2023b) in Domain Adaptation (DA) community that aims to study the cross-domain adaptability of the detection models, we introduce the first 3D reconstruction-simulation dataset, which is constructed based on the Waymo sequences (Sun et al., 2020), with different sensor settings.

We follow the DA community and also employ Waymo (Sun et al., 2020) dataset as the source (old) domain and other datasets, *e.g*, nuScenes (Caesar et al., 2020) and KITTI (Geiger et al., 2012), as the target (new) domain. As a result, the implicit reconstruction is performed on Waymo in order to generate the 3D scene-level meshes, while we simulate KITTI, nuScenes, and ONCE scenarios according to the Waymo-based 3D meshes. The detailed implementation is elaborated in Sec. 4. Besides, Waymo sensor features one top LiDAR and four side LiDARs (Sun et al., 2020), which facilitate a broader longitudinal perception range capable of encompassing the narrow longitudinal field of view of other datasets such as KITTI (Geiger et al., 2012).

**3D Reconstructed Meshes and Simulated Points.** We generate scene-level 3D meshes from Waymo dataset. Note that during the evaluation stage of reconstructed 3D meshes, we can perform the re-raycasting according to Eq. 3 on Waymo domain, and select 146 meshes with the highest reconstruction scores calculated using the Chamfer Distance (CD) distance between the simulated and real points. For cross-dataset setting, we simulate approximately $26K \sim 29K$ samples per domain. The visualization results of the simulation data are illustrated in Fig. 2 and Appendix A.

**Reconstruction-Simulation Dataset Analyses.** Considering that the generated point clouds not only need to boost the model performance in the target domain, but also valuable for 3D pre-training

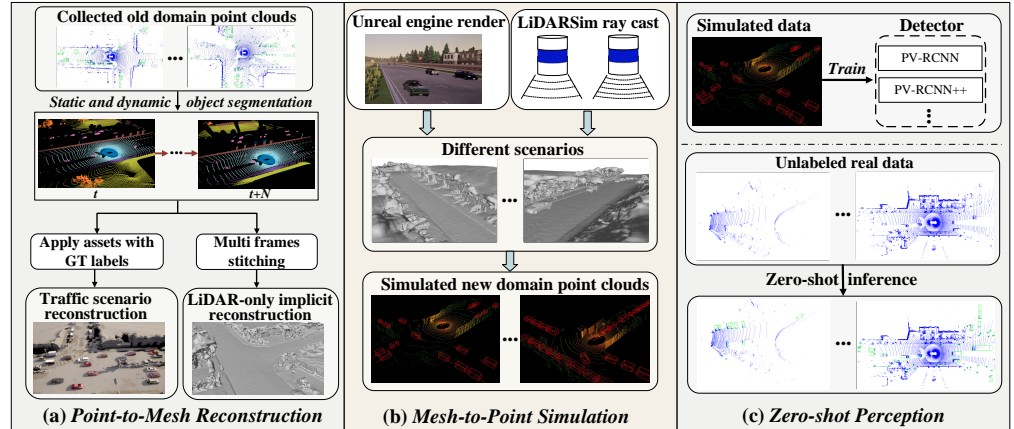

Figure 3: The overview of ReSimAD, which consists of point-to-mesh reconstruction, mesh-to-point simulation, and zero-shot perception. Each part is detailed in Sec. 4.

to enhance the backbone feature generalization ability, we simulate the target-domain points with more diversity of object size. The distribution of object size on the four simulated domains are shown in Fig. 4. It can be seen from Fig 4 that **the reconstruction-simulation dataset** covers a wide object size distribution compared with the off-the-shelf **public** dataset such as ONCE (Mao et al., 2021), Waymo (Sun et al., 2020).

## 4 ReSimAD: Reconstruction, Simulation, Perception Pipeline

**ReSimAD Overview.** As illustrated in Fig. 3, ReSimAD covers three steps.

*1) Point-to-mesh Implicit Reconstruction:* This step aims to map points from the source domain to implicit geometry. To obtain highly consistent simulation data conforming to the distribution of target-domain LiDAR, we reconstruct the real-yet-diverse street scene background, dynamic traffic flow information using LiDAR-only reconstruction (Guo et al., 2023)[1]. We select Waymo as the source domain for scenario reconstruction.

*2) Mesh-to-point Rendering:* The purpose of this step is to simulate the target-domain-like points given the reconstructed implicit geometry, by changing the LiDAR-sensor and scene layout. Specifically, we employ PCSim (Cai et al., 2023)[2] and reproduce the sensor configuration scheme used in the target domain, including LiDAR scan modes, physical characteristics, and deployment locations.

*3) Zero-shot Perception Process:* The well-simulated points are fed into the perception module that can help the original model enhance the cross-domain generalization for common domain variations such as changes in LiDAR types.

*1) Point-to-mesh Implicit Reconstruction.* Inspired by previous works such as StreetSurf, DeepSDF and NeuS (Guo et al., 2023; Park et al., 2019; Wang et al., 2021), we utilize the Implicit Neural Reconstruction method. By leveraging neural networks to encode signed distance functions (Park et al., 2019; Wang et al., 2021; Oechsle et al., 2021; Zhu et al., 2022), we achieve the synthesis of high-quality 3D models with exceptional resolution and efficient memory usage.

Different from recent methods (Deng et al., 2022; Rematas et al., 2022) that use RGB images to refine implicit representation, we carry out **LiDAR-only setting**, which takes sparse LiDAR point clouds as input and generates an implicit Signed Distance Field (SDF) field. LiDAR-based Implicit Neural Reconstruction (LINR) allows us to be unaffected by lighting conditions, *e.g.* weak lighting and changing weather in practical scenes, and to obtain a richer and larger dataset of the 3D scene.

For a ray $r(o, d)$ emitted from the origin $o \in \mathbb{R}^3$ with the direction $d \in \mathbb{R}^3$, we apply volume rendering for LiDAR to train the SDF network, the rendered depth $\widehat{D}$ can be formulated as:

$$\widehat{D}(\mathbf{r}) = \sum_{i=1}^{k} T_i \alpha_i t_i, \tag{1}$$

---

[1] Code of implicit reconstruction is available at: https://github.com/pjlab-ADG/neuralsim
[2] Code of mesh-to-point rendering is available at: https://github.com/PJLab-ADG/PCSim

where $t_i$ is the depth of the $i$-th sampling point, $T_i = \prod_{j=1}^{i-1} (1 - \alpha_j)$ is the accumulated transmittance, and $\alpha_i$ is obtained by employing the close-range model in NeuS (Wang et al., 2021).

Drawing inspiration from StreetSurf (Guo et al., 2023), the reconstruction input is derived from LiDAR rays, and the output is the predicted depth. On each sampled LiDAR beam $\mathbf{r}_{\text{lidar}}$, we apply a logarithm L1 loss on $\widehat{D}^{(\text{cr,dv})}$, rendered depth of the combined close-range and distant-view model:

$$\mathcal{L}_{\text{geometry}} = \ln \left( \left| \widehat{D}^{(\text{cr,dv})} \left( \mathbf{r}_{\text{lidar}} \right) - D \left( \mathbf{r}_{\text{lidar}} \right) \right| + 1 \right). \tag{2}$$

However, the LINR method still faces some challenges. A single LiDAR point cloud frame captures only a portion of the comprehensive information contained within a standard RGB image, due to the inherent sparsity of LiDAR data. This disparity underscores the potential inadequacy of depth rendering in providing the necessary geometric details for effective training. Consequently, this could lead to the generation of a substantial volume of artifacts within the resulting reconstructed mesh. To tackle this challenge, **we consolidate all frames** from the corresponding sequence within Waymo dataset (Sun et al., 2020). Please refer to Appendix A.1 for more details of point cloud registration when consolidating all frames. Next, we use point neighborhood statistics to filter out the outlier point clouds for each scenario.

Considering the constraint of top LiDAR's vertical field of view in Waymo dataset, obtaining point clouds only between -17.6° and 2.4° imposes limitations on the reconstruction of tall surrounding buildings. To tackle this challenge, we introduce a solution that incorporates point clouds from side LiDAR (blind compensation) into the sampling sequence. Four side LiDARs are strategically positioned on the front, rear, and sides of the vehicle, with a vertical field of view spanning from -90° to 30°. This effectively compensates for the inadequate field of view of the top LiDAR. Since the disparity in point cloud density between the side LiDAR and top LiDAR, we opt to assign higher sampling weights to the side LiDAR to enhance the quality of scene reconstruction for tall buildings.

After reconstructing the implicit surface, we can obtain continuous representations of scene geometry with finer granularity, facilitating the extraction of high-resolution meshes for subsequent rendering in a selected simulator. For more details on the training settings of LINR, please refer to StreetSurf (Guo et al., 2023).

**Reconstruction Evaluation.** Due to the occlusion caused by dynamic objects and the influence of LiDAR noise, implicit representations might be lower than expected, posing challenges for cross-domain adaptation. Hence, we conduct an evaluation for the reconstruction accuracy. Since we can access the real-world point clouds of the old domain, we evaluate the accuracy of the reconstruction process by re-raycasting the point clouds for **the old domain**.

We use a set of metrics for the reconstruction accuracy between the rendered points $\widehat{G}$ and original collected LiDAR points $G$, with Root Mean Square Error (RMSE) and Chamfer Distance (CD):

$$\text{CD}(\widehat{G}, G) = \frac{1}{|\widehat{G}|} \sum_{\mathbf{x} \in \widehat{G}} \min_{\mathbf{y} \in G} \|\mathbf{x} - \mathbf{y}\|_2^2 + \frac{1}{|G|} \sum_{\mathbf{y} \in G} \min_{\mathbf{x} \in \widehat{G}} \|\mathbf{y} - \mathbf{x}\|_2^2, \tag{3}$$

where the reconstructed scores and some detailed descriptions are shown in Table 6 in Appendix.

*2) Mesh-to-point Rendering.* After obtaining the static background mesh through the above-mentioned LINR, we use the Blender Python API to convert the mesh data from `.ply` format to 3D model files using `.fbx` format, and finally load the background mesh as an asset into CARLA (Dosovitskiy et al., 2017), an open-source simulator for autonomous driving research.

For the appearance match of traffic participants, we obtain the categories and the three-dimensional size of the bounding boxes in each frame of data, through the annotations from Waymo (Sun et al., 2020). According to this information, we search for the digital asset with the closest size among the traffic participants of the same category in the digital asset library of CARLA (Dosovitskiy et al., 2017), and use it as the model of the traffic participant. According to the scene truth information available in the CARLA simulator, we developed **a bounding box extraction tool** for each detectable target in the traffic scene, and converted it into the label format of the target domain (such as KITTI (Geiger et al., 2012)).

It can be observed from Fig. 4 that the distribution of object sizes is different across datasets. To ensure the consistency between the simulation dataset and the general vehicle size in the target domain, we first perform function mapping on the size of each traffic participant according to the statistical results, and then complete the assets matching process.

For motion simulation of traffic participants, we sort out the traffic scene coordinate system and update the location and pose of dynamic objects frame by frame. For each segment, we take the grounding point of the ego vehicle center in the first frame as the coordinate origin. The 6D pose of the ego vehicle is updated via the difference between ego vehicle labels in different frames. Other dynamic targets are updated by the relative 6D pose of the ego vehicle in the label information of each frame. The 6D pose of a simulated object $P^t$ in the $t$-th frame can be

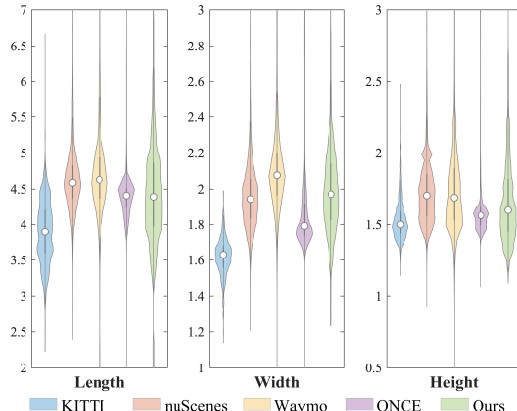

Figure 4: Distribution differences of object size (Length, Width, and Height) across datasets. Compared with the off-the-shelf public datasets, the simulation dataset constructed by the proposed ReSimAD covers a wider distribution.

represented as $(x, y, z, roll, yaw, pitch)$ in the simulator. The update of the ego pose and dynamic object pose is:

$$P_{ego}^t = L_{ego}^t - L_{ego}^0, \quad P_{target}^t = L'^t_{target} + P_{ego}^t, \tag{4}$$

where $L$ denotes the absolute pose in Waymo world coordinate system, and $L'$ is the relative pose in the coordinate system relative to the ego pose.

To study the effects of traffic scenario reconstruction and LiDAR simulation on synthetic data authenticity and zero-shot domain adaptation performance, three datasets are constructed. In addition to the implicit reconstruction simulation dataset according to the aforementioned method, the sensor-like LiDAR simulation dataset and the default LiDAR simulation dataset both based on the CARLA scenario background are also constructed, by the OpenCDA tool (Xu et al., 2021b).

The main difference between the sensor-like LiDAR dataset and the default LiDAR simulation dataset is that, the number of LiDAR channels and the vertical field of view are different. The default LiDAR has a fixed 32-channel configuration with a vertical field of view ranging from -30 to 10 degrees, and beams are uniformly distributed. Meanwhile, leveraging the simulation library (Cai et al., 2023), the features of the sensor-like LiDAR are equal to those of the corresponding sensor setting from the target domain. The detection range, points emitted per second, rotation frequency, and drop-rate of the target domain LiDAR are also nearly identical to those of the default LiDAR. Due to the need for vehicle traffic flow to match the road network structure, for data simulation based on the CARLA static background, we completed the matching of vehicle traffic flow.

***3) Zero-shot Perception Process.*** To further achieve the closed-loop simulation verification, we use the simulated points $X_{sim}$ to train our baseline model on the new domain, and perform an evaluation on the real samples $X_{real}$ of the validation set from the new domain. Specifically, we verify our method on the 3D detection task, and the overall loss $L_{train}$ and evaluation process $E_{eval}$ are:

$$\mathcal{L}_{train} = \mathcal{L}_{cls}(X_{sim}) + \mathcal{L}_{reg}(X_{sim}), \quad E_{eval} = AP(\hat{X}_{real}, X_{real}^{GT}), \tag{5}$$

where $\mathcal{L}_{cls}$ and $\mathcal{L}_{reg}$ are the class and regression loss of the detection head, respectively. $\hat{X}_{real}$ is the prediction. $AP$ is the metric score between the model prediction and ground truth (see Sec. 5.1).

## 5 EXPERIMENTS

### 5.1 EXPERIMENTAL SETUP

**Dataset Selection for Experiments.** In order to make fair comparisons with the existing UDA methods (Yang et al., 2021; 2022; Wei et al., 2022; Yuan et al., 2023b), we align their cross-dataset setting, and employ the Waymo-to-KITTI, Waymo-to-nuScenes setting to study the model cross-domain ability under the 3D scenario. Besides, to achieve high-quality reconstruction results, we

Table 1: Results on different adaptation scenarios under zero-shot, Unsupervised Domain Adaptation (UDA), and Fully-supervised Training (FT) settings. Following (Yang et al., 2021), we report $AP_{BEV}$ and $AP_{3D}$ over 40 positions' recall for car category. **Source Only** denotes that the pre-trained detector is directly evaluated on the target domain, and **Oracle** represents the detection results trained using the fully-annotated target domain. Closed Gap denotes the detection accuracy gap closed by various methods along Source Only and Oracle results. Our method belongs to zero-shot, while UDA assumes that all REAL target samples are available for model training. The best results of model adaptation are marked in **bold**.

| Task | Method | Setting | PV-RCNN | | PV-RCNN++ | |
|---|---|---|---|---|---|---|
| | | | $AP_{BEV}$ / $AP_{3D}$ | Closed Gap | $AP_{BEV}$ / $AP_{3D}$ | Closed Gap |
| Waymo→KITTI | Source Only | Zero-shot | 61.18 / 22.01 | - | 67.68 / 20.81 | - |
| | ST3D (Yang et al., 2021) | UDA | **84.10** / **64.78** | +82.45% / +70.71% | 62.55 / 17.53 | -38.16% / -6.53% |
| | ReSimAD (Ours) | Zero-shot | 81.01 / 58.42 | +71.33% / +60.19% | **82.06** / **61.65** | +106.99% / +81.32% |
| | CARLA-default (Dosovitskiy et al., 2017) | Zero-shot | 58.69 / 34.21 | -0.09% / +20.16% | 50.65 / 31.95 | -126.71% / +22.18% |
| | Sensor-like (Manivasagam et al., 2020) | Zero-shot | 71.09 / 40.80 | +35.64% / +31.06% | 53.16 / 34.16 | -108.03% / +26.58% |
| | Oracle | FT | 88.98 / 82.50 | - | 81.12 / 71.03 | - |
| Waymo→nuScenes | Source Only | Zero-shot | 31.02 / 17.75 | - | 29.93 / 18.77 | - |
| | ST3D (Yang et al., 2021) | UDA | 36.42 / **22.99** | +24.44% / +25.18% | 34.68 / 17.17 | +19.40% / -7.92% |
| | ReSimAD (Ours) | Zero-shot | **37.85** / 21.33 | +30.92% / +17.20% | **40.73** / **23.72** | +44.12% / +24.52% |
| | CARLA-default (Dosovitskiy et al., 2017) | Zero-shot | 24.14 / 12.26 | -31.14% / -26.38% | 22.14 / 7.04 | -31.82% / -58.09% |
| | Sensor-like (Manivasagam et al., 2020) | Zero-shot | 28.90 / 15.35 | -9.59% / -11.53% | 35.98 / 19.57 | +24.71% / +3.96% |
| | Oracle | FT | 53.11 / 38.56 | - | 54.41 / 38.96 | - |
| Waymo→ONCE | Source Only | Zero-shot | 68.82 / 39.06 | - | 68.72 / 34.39 | - |
| | ST3D (Yang et al., 2021) | UDA | 68.13 / 41.53 | -3.43% / +6.04% | 70.81 / 36.79 | +10.74% / +5.34% |
| | ReSimAD (Ours) | Zero-shot | **70.97** / **48.59** | +10.68% / +23.30% | **74.52** / **53.91** | +29.82% / +43.43% |
| | CARLA-default (Dosovitskiy et al., 2017) | Zero-shot | 58.82 / 32.11 | -55.40% / -16.99% | 56.80 / 32.77 | -61.29% / -3.60% |
| | Sensor-like (Manivasagam et al., 2020) | Zero-shot | 61.38 / 35.77 | -36.96% / -8.04% | 69.13 / 44.16 | +2.11% / +21.74% |
| | Oracle | FT | 88.95 / 79.97 | - | 88.17 / 79.34 | - |

have to merge multiple frames sampled from the same 3D sequence to perform the reconstruction process, which is also inspired by StreetSurf (Guo et al., 2023).

**Implementation.** We first train the base model on the labeled source domain and evaluate the cross-domain performance of the trained source model on the target domain. We define the **domain variations** in different cases including cross-region, cross-beam, and cross-dataset, *etc*. For training on the simulated data, we use Adam optimizer with a one-cycle learning rate decay schedule, where we set the maximum learning rate to 0.002. The weight decay is set to 0.01. The total simulation data-related training process ends when 10 epochs are reached using 4 NVIDIA Tesla A100 GPUs.

**Evaluation Metric.** Following previous cross-domain research works (Yang et al., 2021; 2022; Wei et al., 2022; Yuan et al., 2023b; Zhang et al., 2023), we use AP for evaluation in both the Bird's Eye View (BEV) and 3D over 40 recall positions, where the IoU threshold is set to 0.7.

**Comparison Baselines.** We compare the proposed ReSimAD with three typical baselines of alleviating cross-domain shifts: 1) Data simulation-related baseline using simulation engine (Dosovitskiy et al., 2017); 2) Sensor simulation-related baseline by changing the sensor parameter setting; 3) Label-efficient baseline (Yang et al., 2021).

**1) CARLA-default**: We use the CARLA (Dosovitskiy et al., 2017) simulation engine to generate the simulated data. For this baseline, foreground objects are added into the simulated scenario by searching from CARLA to find the closest object-size digital asset, as described in Sec. 4.

**2) Sensor-like**: We assume that the target-domain sensor setting can be obtained, and thus, we also change the sensor parameter setting in CARLA and simulate point data so that the LiDAR-beam distribution is similar to the target domain scene. **Note that** the above two baseline settings only produce more simulated data that conform to the target-domain distribution, and we directly use the simulated data to fine-tune our base models and observe their performance in the target domain.

**3) ST3D**: We compare with ST3D (Yang et al., 2021), a popular Unsupervised Domain Adaptation (UDA) technique reducing the cross-domain differences of point clouds in a label-efficient manner.

### 5.2 CROSS-DOMAIN EXPERIMENTS

#### 5.2.1 ReSimAD BOOSTS ZERO-SHOT 3D OBJECT DETECTION

To ensure the fairness of experiments, we first compare the proposed ReSimAD with the data simulation-related baselines: CARLA-default (Dosovitskiy et al., 2017) and Sensor-like (Manivasagam et al., 2020). It can be observed from Table 1 that, our ReSimAD achieves the best zero-shot 3D detection accuracy for **all cross-domain settings** on both PV-RCNN (Shi et al., 2020) and PV-RCNN++ (Shi et al., 2021). Also, we found that the Sensor-like baseline (Manivasagam et al., 2020) has stronger robustness against domain differences compared with CARLA-default (Doso-

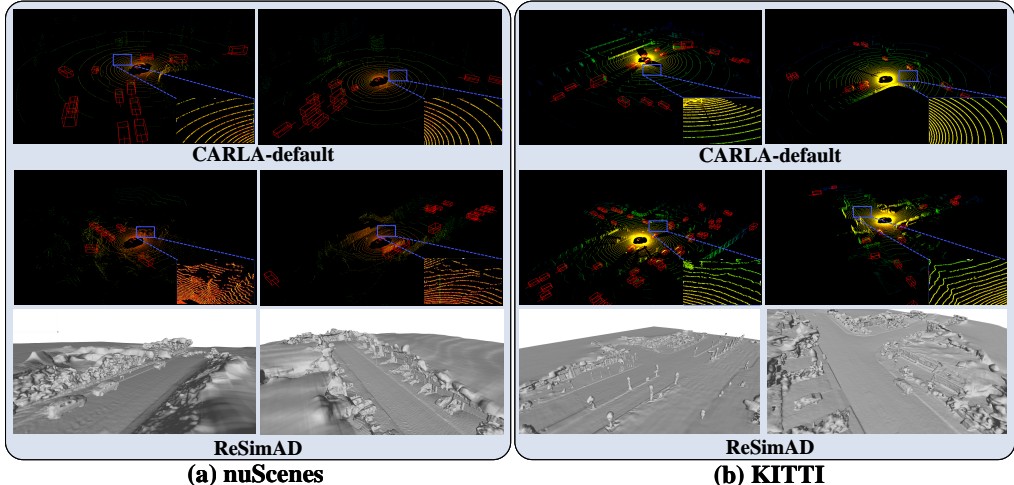

Figure 5: Visualization of simulated points using CARLA-default background (first row) or reconstructed one (second row) from the real scenarios. The last row: reconstructed background.

vitskiy et al., 2017), since we conduct sensor-level simulation in advance according to the LiDAR parameter settings of the target domain. However, due to the differences in background distribution between simulated and real scenes, it is still difficult to achieve satisfactory cross-domain performance using Sensor-like baseline (Manivasagam et al., 2020) alone (*i.e.*, only 71.09%/40.80% for Waymo-to-KITTI setting).

Besides, Table 1 compares Sensor-like (Manivasagam et al., 2020) and ReSimAD, and it shows that ReSimAD can generally outperform the Sensor-like methods by around $5.98\% \sim 27.49\%$ under different types of cross-domain differences. Therefore, we believe that the authenticity of the **point-cloud background** distribution is also crucial for achieving zero-shot cross-dataset detection.

Table 1 shows the results of leveraging Unsupervised Domain Adaptation (UDA) technique. The major difference between UDA and ReSimAD is that, the former employs samples from **real scenes** of the target domain for model adaptation, while the latter **cannot** access any real point cloud data from the target domain. From Table 1, the cross-domain results achieved by our ReSimAD are comparable to that achieved by UDA method (Yang et al., 2021). This result indicates that our method can greatly reduce the cost of data acquisition, and further, shorten the development cycle of model retraining when the LiDAR sensor needs to be upgraded.

Table 2: Zero-shot and Fully-supervised (SFT and Oracle) results on the target domain. For SFT setting, we use the checkpoint pre-trained on the simulated data as the backbone initialization, and fine-tune on the labeled target domain.

| Models | Setting | Waymo→KITTI $AP_{BEV}$ / $AP_{3D}$ | Waymo→nuScenes $AP_{BEV}$ / $AP_{3D}$ |
|---|---|---|---|
| PV-RCNN (Shi et al., 2020) | Zero-shot | 81.01 / 58.42 | 37.85 / 21.33 |
| PV-RCNN (Shi et al., 2020) | SFT | **88.30** / **82.71** | **54.48** / **38.72** |
| PV-RCNN (Shi et al., 2020) | Oracle | 88.03 / 81.61 | 53.07 / 38.39 |
| PV-RCNN++ (Shi et al., 2021) | Zero-shot | 82.06 / 61.65 | 40.73 / 23.72 |
| PV-RCNN++ (Shi et al., 2021) | SFT | **87.95** / **81.55** | **55.52** / **39.94** |
| PV-RCNN++ (Shi et al., 2021) | Oracle | 86.39 / 80.24 | 54.41 / 38.96 |

### 5.2.2 ReSimAD Boosts Fully-supervised 3D Detection

Another benefit of using data generated by ReSimAD is that a high-performance target-domain accuracy can be obtained without accessing any target-domain real data distribution. We found that such a target-domain-like simulation process can further boost the Oracle results of the baseline.

Table 2 reports the results of using full human annotations from the target domain. Oracle represents the highest result achieved by the baseline model trained on all labeled data from the target domain. SFT denotes that the network parameter of the baseline model is initialized by the weights trained from the simulation data. Table 2 shows that the backbone pre-trained using our simulated point clouds provides a better initialization for 3D detectors such as PV-RCNN++ and PV-RCNN.

### 5.3 3D Pre-training Experiments

**Overview of 3D Pre-training using Simulation Data.** To verify if ReSimAD can produce point data to help 3D pre-training, we devise the following setting: 3D backbone is pre-trained on **the simulated point clouds** using AD-PT (Yuan et al., 2023a), and then fine-tuned on the downstream real-world data. It saves lots of real-world data by using simulated data for 3D pre-training.

Table 3: The performance scalability over different amounts of training data, using the simulated dataset compared with the real dataset (ONCE). FT denotes the fine-tuning. Sim-data indicates the simulated point data using our method, and ONCE-data are collected in the real-world scene.

| Pre-training Methods | Pre-training Data Source | Data Collection Cost | FT to KITTI $AP_{BEV}$ / $AP_{3D}$ | | | | FT to Waymo | |
|---|---|---|---|---|---|---|---|---|
| | | | Overall | Car | Pedestrian | Cyclist | AP | APH |
| From Scratch (w/o pre-training) | None | None | 70.57 | 84.50 | 57.06 | 70.14 | 69.97 | 67.58 |
| AD-PT (Yuan et al., 2023a) | 55K Sim-data | None | 71.06 | 84.53 | 57.02 | 71.65 | 70.23 | 67.87 |
| AD-PT (Yuan et al., 2023a) | 110K Sim-data | None | 71.74 | **84.82** | 58.30 | 72.10 | 70.46 | 68.42 |
| AD-PT (Yuan et al., 2023a) | 100K ONCE-data | High | **73.01** | 84.75 | **60.79** | **73.49** | **71.55** | **69.23** |

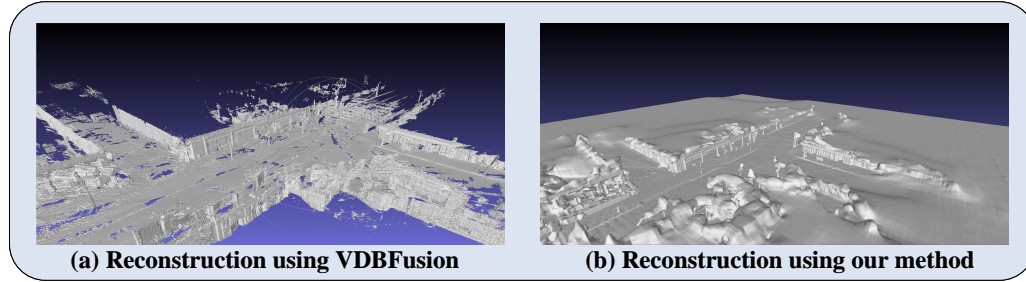

(a) Reconstruction using VDBFusion    (b) Reconstruction using our method

Figure 6: Visualization comparisons between the VDBFusion (Vizzo et al., 2022) and our reconstruction method towards a sequence on Waymo dataset.

**Downstream Fine-tuning Results.** We utilize ReSimAD to generate the data with more wider distribution of point clouds. For a fair comparison with their pre-training results in AD-PT (Yuan et al., 2023a), the target amount of simulation data generated by ReSimAD is approximately 100K. In Table 3, the baseline detector is pre-trained on either simulated data or real-world data, and fine-tuned on both KITTI (Geiger et al., 2012) and Waymo (Sun et al., 2020) benchmarks. Table 3 shows that, the performance of fine-tuning can be continuously increased using simulation pre-training data at different scales. Overall, it achieves pre-training by leveraging different scales of simulated point clouds in a zero-shot fashion.

## 5.4 FURTHER ANALYSES

**Effectiveness of Reconstruction and Simulation.** To verify the module-wise effectiveness of the proposed method, we visualize the point clouds rendered using different methods including CARLA simulator, and real-world reconstructed 3D scenario by our method in Fig. 5.

It shows that the simulated points obtained by ReSimAD cover more realistic scene information for the target domain, such as road surface and streetscape. Fig. 6 also visualizes the reconstructed mesh using different reconstruction methods. Visualization results illustrate that compared with the VDBFusion (Vizzo et al., 2022), the implicitly reconstructed meshes by our ReSimAD show clear street view information and continuous geometric structure. Please refer to Appendix A.4 for more comparison results.

Table 4: Cross-dataset results using SECOND-IOU baseline, where #Sim-data denotes the number of simulated target-domain samples.

| Models | Setting | #Sim-data | Waymo→nuScenes $AP_{BEV}$ / $AP_{3D}$ |
|---|---|---|---|
| SECOND-IOU | Zero-shot | 0 | 24.57 / 15.12 |
| SECOND-IOU | Zero-shot | 5k | 30.12 / 9.27 |
| SECOND-IOU | Zero-shot | 10K | 35.38 / 18.55 |
| SECOND-IOU | Zero-shot | 20K | **36.45 / 18.94** |
| SECOND-IOU | Oracle | 0 | 50.54 / 33.41 |

**Adaptability on Lightweight Baseline.** Table 4 shows that the simulated point clouds are also effective for the lightweight baseline, which is more practical in real applications. We employ SECOND-IOU (Yan et al., 2018) as the baseline, and train it on the simulated data using ReSimAD. The results show that our method also obtains promising results on the one-stage 3D detector.

## 6 CONCLUSION

In this work, we study how to achieve a zero-shot domain transfer, and present ReSimAD consisting of a real-world point-level implicit reconstruction process and a mesh-to-point rendering process. We have conducted extensive experiments under zero-shot settings, and their results demonstrate the effectiveness of ReSimAD in producing target-domain-like samples and achieving high target-domain perception ability, even helpful for 3D pre-training.

ETHICS STATEMENT

The proposed ReSimAD tries to achieve zero-shot domain transfer from a new perspective: source-domain reconstruction and target-domain simulation. Such a zero-shot method can significantly reduce the cost of both data acquisition and human annotation on the target domain. However, due to that knowledge transfer from label-rich source domain to an unseen target domain may contain deviations, the adapted 3D model may produce certain prediction errors towards the target domain. Thus, how to obtain a target-domain model with high performance under the zero-shot setting needs to be further studied in the future.

REPRODUCIBILITY STATEMENT

The complete reconstruction simulation pipeline for ReSimAD has been included in Sec. 4, and the Reconstruction-Simulation Dataset introduced in Sec. 3 has been released. Besides, the source code of ReSimAD is available at: https://github.com/PJLab-ADG/3DTrans.

ACKNOWLEDGEMENT

The research was supported by the National Key R&D Program of China (Grant No. 2022ZD0160104), the Science and Technology Commission of Shanghai Municipality (Grant No. 22DZ1100102), and Shanghai Rising Star Program (Grant No. 23QD1401000).

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

# A APPENDIX

## A.1 POINT-TO-MESH DETAILED IMPLEMENTATION

**Source Domain Dataset Selection.** For selecting the dataset for source domain reconstruction, our primary focus is on whether the LiDAR has a sufficiently large vertical Field of View (vFoV), at least equal to or larger than that of the target domain. If the source domain cannot capture information from a broader area, it will be incapable of acquiring reconstruction information related to the target domain's range. This limitation manifests in the reconstruction process as artifacts or voids appearing in the grid in areas beyond the point cloud FoV of the source domain or in distant regions. To avoid incorporating these parts of the region into the target domain data during the LiDAR rendering process, we ensure that the target domain's vFoV is less than or equal to that of the source domain's LiDAR.

The parameters of the LiDAR systems in Waymo (Sun et al., 2020), nuScenes (Caesar et al., 2020), and KITTI (Geiger et al., 2012) are as shown in Table 5:

Table 5: Vertical Field of View (vFoV) ranges for various datasets.

| Dataset | | vFoV |
|---|---|---|
| Waymo | *top* | [-17.6, 2.4] |
| | *side × 4* | [-90, 30] |
| KITTI | | [-24.9, 2] |
| nuScenes | | [-30.67, 10.67] |

**SDF Reconstruction.** We show some details of point-to-mesh implicit reconstruction. We employ a Signed Distance Field (SDF) to implicitly encode the geometry of street scene objects from LiDAR point cloud data by training a multi-layer fully-connected neural network to fit the target SDF. Specifically, the surface $S$ of the object is represented by the zero-level set of its SDF as follows:

$$\mathcal{S} = \left\{ \mathbf{x} \in \mathbb{R}^3 \mid SDF(\mathbf{x}) = 0 \right\}, \tag{6}$$

where the SDF values $\widehat{S}_i$ are mapped to $\alpha_i$ for volume rendering as follows:

$$\alpha_i = \max \left( \frac{\Phi_s\left(\widehat{\mathcal{S}}_i\right) - \Phi_s\left(\widehat{\mathcal{S}}_{i+1}\right)}{\Phi_s\left(\widehat{\mathcal{S}}_i\right)}, 0 \right), \tag{7}$$

where $s$ is a learnable scaling parameter of the Sigmoid function $\Phi_s(x) = \left(1 + e^{-s \cdot x}\right)^{-1}$. This mapping strategy ensures unbiased calculation of color contribution (*i.e.*, visibility weights) while respecting occlusion.

**The Computational Requirement for Mesh Reconstruction.** For each scene, we train LINR from scratch without any pre-training. Under the LiDAR-only setting, the computational cost required for each sequence's reconstruction task is:

- $\leq 2$ hrs training time on a single RTX3090
- about 16 GiB GPU Memory
- $> 24$ GiB CPU Memory (caching data to speed up)

**Point Cloud Registration.** The Waymo dataset provides precise vehicle and LiDAR poses for every frame in all sequences. It is worth mentioning that, to avoid motion blur, before consolidating all frames, we first remove dynamic objects from the point cloud and only stitch together multiple frames of the static background. We further filter out outlier point clouds in the scene using point neighborhood statistics. Subsequently, we align the point cloud data from each frame in the current sequence to the starting frame of the sequence by projecting them using the LiDAR poses, thus achieving point cloud registration.

**Traffic Flow Reconstruction.** Due to the strong correlation between vehicle movement and road network structure, vehicles generally cannot travel to non-driving areas. After we complete the

Table 6: The sorting results of reconstruction scores (97%) using Root Mean Square Error (RMSE) and Chamfer Distance (CD).

| Sequence Name | RMSE | CD Distance |
|---|---|---|
| seg2752216 | 0.0434 | 0.0050 |
| seg3451017 | 0.0590 | 0.0152 |
| seg1066482 | 0.0596 | 0.0111 |
| seg6343780 | 0.0659 | 0.0288 |
| seg1125208 | 0.0672 | 0.0177 |
| ... | ... | ... |
| seg1255991 | 6.3363 | 11.7660 |
| seg1048592 | 7.3421 | 42.9982 |
| seg1585730 | 8.4834 | 127.2535 |

background reconstruction including roads with the source domain dataset, if the matching of vehicle traffic flow cannot be ensured in rendering, it may result in abnormal vehicle movement, such as collisions with surrounding buildings. Therefore, it is crucial to ensure the same density of vehicle traffic flow as the source domain dataset. With the information of each target object the position update in the source domain, we simulate the traffic flow in CARLA.

**Reconstruction Metric.** Root Mean Square Error (RMSE) we used for evaluating reconstruction meshes can be formulated as follows Godard et al. (2019):

$$RMSE = \sqrt{\frac{\sum_i^N (\widehat{D}_i - D_i)}{N}}, \tag{8}$$

where $\widehat{D}_i$ represents the rendered depth, and $D_i$ is the ground truth depth obtained from LiDAR.

When computing Chamfer Distance (CD), we consider the upper 97% of effective points to determine the average value of the CD distance, while disregarding outliers characterized by significant errors caused by complicated 3D scenes.

## A.2 LIDAR RAY CAST PRINCIPLE

Utilizing ray cast tracing to emulate the Time of Flight (ToF) technique (Royo & Ballesta-Garcia, 2019), CARLA generates authentic synthetic point clouds by simulating the interaction between rays and object collisions. The LiDAR sensor employs laser emissions to scan a specified region, following a predetermined angular increment across a spherical plane. The orientation of the laser is governed by azimuth angle $\theta$ and polar angle $\phi$. By extracting the distance $r$ through ray cast operations, the coordinates of point $p_i$ can be computed as follows:

$$p_i = \begin{pmatrix} x_i \\ y_i \\ z_i \end{pmatrix} = r_i \begin{pmatrix} \cos\theta_i \cos\phi_i \\ \cos\theta_i \sin\phi_i \\ \sin\theta_i \end{pmatrix}. \tag{9}$$

We utilize the official specifications of the LiDAR used in the target domain to complete the modeling of scanning characteristics, while also considering the simulation of physical properties. The key technical details of this part are fully elaborated in previous work on LiDAR simulation (Cai et al., 2023).

## A.3 DATASET DESCRIPTION

**Waymo Open Dataset.** Waymo Open Dataset (Sun et al., 2020), a large-scale autonomous driving dataset collected using 64-beam LiDAR sensor, covers a total of 1150 scene sequences, which are further divided into a train set with 798 sequences, a validation set with 202 sequences, and a test set with 150 sequences. Each sequence spans approximately 20 seconds and contains 200 frames of point clouds.

**nuScenes Dataset.** nuScenes (Caesar et al., 2020) collects point clouds using 32-beam LiDAR sensor, consisting of 28130 training samples and 6019 validation samples. Besides, it encompasses 1000 driving scenarios collected in both Boston and Singapore.

**KITTI Dataset.** KITTI (Geiger et al., 2012), collected in Germany, contains point cloud data captured by a 64-beam LiDAR. It consists of 7481 training samples and 7581 test samples, where the train set is further divided into 3712 and 3769 samples for training and validation, respectively.

**ONCE Dataset.** ONCE (Mao et al., 2021) is a large-scale autonomous dataset collected in China using 40-beam LiDAR sensor. It encompasses a diverse range of data collected at various times, under different weather conditions, and across multiple regions. The dataset covers massive unlabeled point clouds (over 1M frames of point cloud data) and approximately 15K labeled point clouds.

### A.4 MORE VISUALIZATION RESULTS

Here, we illustrate more visualization results of the proposed simulation dataset and the reconstruction meshes in Figs. 7 and 8. Also, by visualizing the results, we can observe the shortcomings of the CARLA simulation engine: lacking the authenticity of outdoor scenes. For example, by comparing the CARLA-default and ReSimAD in Fig. 7, it can be seen that the visualization results present the circular-like simulated points using the CARLA-default method. This is mainly due to that the current simulator is unable to simulate realistic background information such as road structure.

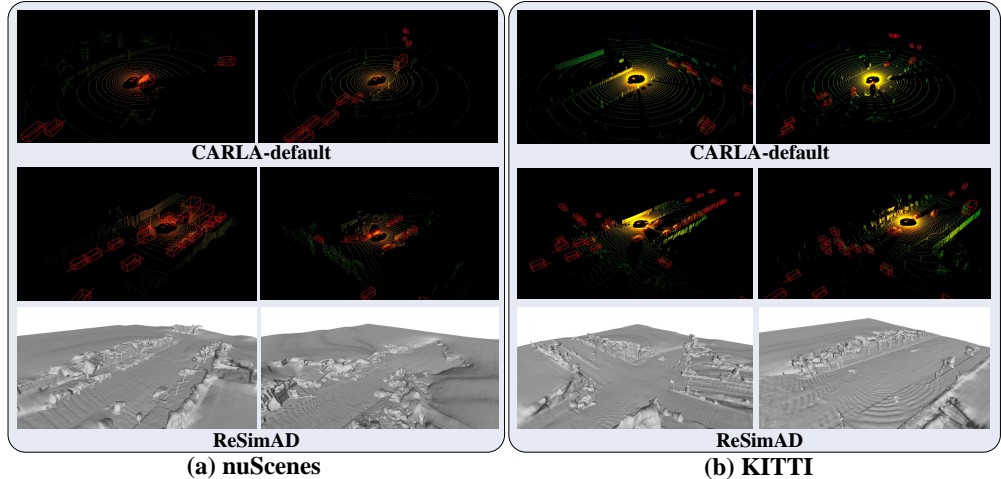

Figure 7: Visualization of simulated points using CARLA-default background or reconstructed background. The 1st and 2nd row indicates the simulated results using CARLA-default background and reconstructed background, respectively. The last row represents the reconstructed background.

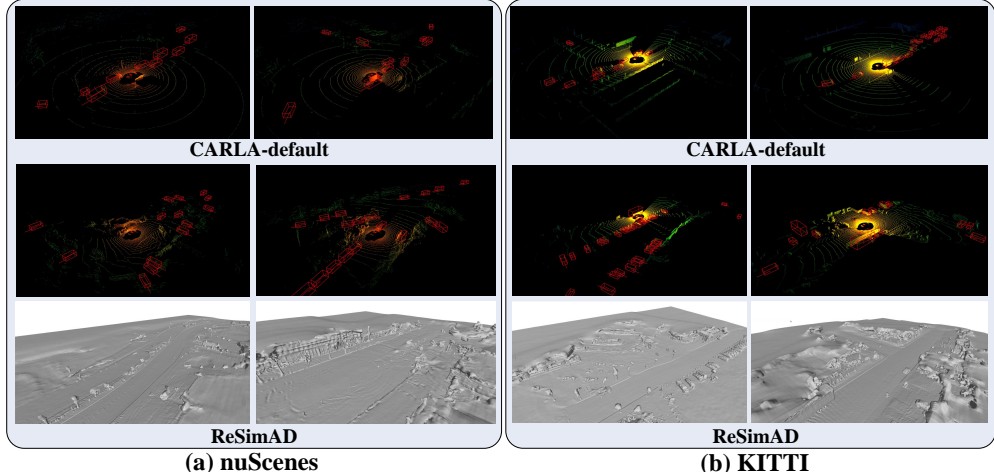

Figure 8: Visualization of simulated points using CARLA-default background or reconstructed background. The 1st and 2nd row indicates the simulated results using CARLA-default background and reconstructed background, respectively. The last row represents the reconstructed background.

