# OpenReview forum: "ReSimAD: Zero-Shot 3D Domain Transfer for Autonomous Driving with Source Reconstruction and Target Simulation"
_ICLR.cc/2024/Conference — ICLR 2024 poster_

### Official Review · Reviewer_Fz1j · 2023-10-27

**Soundness:** 3 good
**Presentation:** 2 fair
**Contribution:** 3 good
**Rating:** 6
**Confidence:** 4

**Summary:**

The paper introduces ReSimAD, a unified reconstruction-simulation-perception paradigm that addresses the domain shift issue when 3D detectors face cross-domain performance degradation in the field of autonomous driving.
The experimental results conducted on multiple 3D detectors and datasets verify that the proposed paradigm can maintain cross-domain performance in the simulated target domain. In addition, it can also assist and accelerate model training optimization, which has a certain application value.

**Strengths:**

Regarding the phenomenon of cross-domain performance degradation in 3D detectors in the field of autonomous driving, this paper proposes a paradigm of reconstruction-simulation-perception from the perspective of data sources. This paradigm can alleviate or partially solve the problem of cross-data domain discrepancies. The proposed approach, named ReSimAD, has significant research value.

**Weaknesses:**

1)	The readability of the paper needs improvement, such as the illustrations, explanations, and better formatting.
2)	Some necessary elaborations and supporting evidence need to be added.

**Questions:**

1)	Figure 1 and 3 can be made more illustrative of the proposed paradigm.
2)	The formatting of the paper needs adjustment, for example, ensure that figures and corresponding sections are not too far apart.
3)	To eliminate artifacts in Point-to-mesh Implicit Reconstruction, it should be further explained how the authors performed point cloud registration when consolidating all frames from the corresponding sequence.
4)	The interpretation of "Closed Gap" in Table 2 and its analysis are not provided in the paper.
5)	The paper does not mention the significance or the specific impact of "the matching of vehicle traffic flow density" mentioned in Section 4 (Mesh-to-point Rendering).
6)	In Table 5, what is the reason of the significant gap between using zero-shot and oracle methods? The details regarding the sample quantity and other experimental settings can be supplemented in the paper.

---

> ### Author Response · Authors · 2023-11-21
> **Response to Reviewer Fz1j**
>
> **Q1**: Figures 1 and 3 can be made more illustrative of the proposed paradigm.
>
> **A1**: We are grateful to the reviewers for pointing out these problems in the manuscript. We have polished Figures 1 and 3 of the manuscript to further clearly our method.
>
> &ensp;
>
> **Q2**: The formatting of the paper needs adjustment, for example, ensure that figures and corresponding sections are not too far apart.
>
> **A2**: We have noticed the placement issue with Tables 2, 3, and 4. Thank you very much for pointing that out, and we will make the necessary corrections in the upcoming updated version.
>
> &ensp;
>
> **Q3**: To eliminate artifacts in Point-to-mesh Implicit Reconstruction, it should be further explained how the authors performed point cloud registration when consolidating all frames from the corresponding sequence.
>
> **A3**: We utilize the well-annotated Waymo dataset, which provides precise vehicle and LiDAR poses for each frame in all sequences. It is worth mentioning that, to avoid motion blur, prior to consolidating all frames, we first remove dynamic objects from the point cloud and only stitch together multiple frames of the static background. We further filter out outlier point clouds in the scene using point neighborhood statistics. Subsequently, we align the point cloud data from each frame in the current sequence to the starting frame of the sequence by projecting them using the LiDAR poses, thus achieving point cloud registration.
>
> &ensp;
>
> **Q4**: The interpretation of "Closed Gap" in Table 2 and its analysis are not provided in the paper.
>
> **A4**:  Thanks. As stated in caption of Table 2 of the manuscript, the "Closed Gap" denotes the detection accuracy drop rate closed by various methods along the Source Only and Oracle results, which is a common evaluation metric used by previous Unsupervised Domain Adaptation (UDA) works [1, 2]. "Closed Gap" can be calculated by: $(AP_{methods} - AP_{Source-only}) / (AP_{Oracle} - AP_{Source-only})$.
>
> **References**:
>
> [1] Jihan Yang, Shaoshuai Shi, Zhe Wang, Hongsheng Li, and Xiaojuan Qi. St3d: Self-training for unsupervised domain adaptation on 3d object detection. In Proceedings of the IEEE/CVF Conference on Computer Vision and Pattern Recognition, pp. 10368–10378, 2021.
>
> [2] Jiakang Yuan, Bo Zhang, Xiangchao Yan, Tao Chen, Botian Shi, Yikang Li, and Yu Qiao. Bi3d: Bi-domain active learning for cross-domain 3d object detection. In Proceedings of the IEEE/CVF Conference on Computer Vision and Pattern Recognition, pp. 15599–15608, 2023b.
>
> &ensp;
>
> **Q5**: The paper does not mention the significance or the specific impact of "the matching of vehicle traffic flow density" mentioned in Section 4 (Mesh-to-point Rendering).
>
> **A5**: Due to the strong correlation between vehicle movement and road network structure, vehicles generally cannot travel to non-driving areas. In this paper, we rely on the Waymo Open Dataset as the source domain dataset to complete background reconstruction, including roads. If the matching of vehicle traffic flow cannot be ensured in rendering, it may result in abnormal vehicle movement, such as collisions with surrounding buildings. Therefore, it is crucial to ensure the same density of vehicle traffic flow as the source domain dataset.
>
> &ensp;
>
> **Q6**: In Table 5, what is the reason of the significant gap between using zero-shot and oracle methods? The details regarding the sample quantity and other experimental settings can be supplemented in the paper.
>
> **A6**: Thanks for your valuable suggestions. In Table 5, the main reason for the significant accuracy gap between using zero-shot and Oracle is that, the Waymo-to-nuScenes cross-dataset setting is very challenging for 3D UDA study, as reported in other works [1, 2, 3], since Waymo and nuScenes datasets have totally different LiDAR beam and scene layout.
>
> **References**:
>
> [1] Jihan Yang, Shaoshuai Shi, Zhe Wang, Hongsheng Li, and Xiaojuan Qi. St3d: Self-training for unsupervised domain adaptation on 3d object detection. In Proceedings of the IEEE/CVF Conference on Computer Vision and Pattern Recognition, pp. 10368–10378, 2021.
>
> [2] Jiakang Yuan, Bo Zhang, Xiangchao Yan, Tao Chen, Botian Shi, Yikang Li, and Yu Qiao. Bi3d: Bi-domain active learning for cross-domain 3d object detection. In Proceedings of the IEEE/CVF Conference on Computer Vision and Pattern Recognition, pp. 15599–15608, 2023b.
>
> [3] Zhuoxiao Chen, Yadan Luo, Zheng Wang, Mahsa Baktashmotlagh, Zi Huang. Revisiting Domain-Adaptive 3D Object Detection by Reliable, Diverse and Class-balanced Pseudo-Labeling. In *Proceedings of the IEEE/CVF International Conference on Computer Vision* (pp. 3714-3726).

---

> > ### Author Response · Authors · 2023-11-23
> > **Manuscript has been updated**
> >
> > Dear Reviewer Fz1j，
> >
> > Thanks for your positive comments and valuable suggestions. According to your suggestions during the previous round, we have further revised our manuscript from the following aspects:
> >
> > - **Enhance readability**: Figures 1 and 3 have been redrawn to clearly illustrate the pipeline of the proposed ReSimAD. Please see Figure 1 on Page 2 and Figure 3 on Page 4. Besides, the formatting of this paper (including Tables 1, 2, and 3, Figures 4 and 5) has been carefully adjusted, in order to ensure that the tables or figures are aligned with the corresponding text. Besides, all the mistakes and errors which we can find have been revised and corrected.
> >
> > - **Necessary elaborations and supporting evidence**: Explanations of point cloud registration when consolidating all frames from the corresponding sequence have been included on Page 5 of the revised manuscript. Besides, the details regarding the sample quantity and other experimental settings have been supplemented on Appendix A.3 due to the page limitation.
> >
> > ALL responses during the rebuttal period have been included in the revised manuscript. It is appreciated that you could consider these modifications and rapid actions of our manuscript.
> >
> > &ensp;
> >
> > Best regards,
> >
> > Authors of Paper 1573.

---

### Official Review · Reviewer_bhEL · 2023-10-31

**Soundness:** 2 fair
**Presentation:** 1 poor
**Contribution:** 2 fair
**Rating:** 5
**Confidence:** 4

**Summary:**

This paper introduces 3d domain transfer pipeline, ReSimAD.
It focuses on bridging the gap between old autonomous driving datasets and newly collected real-life datasets.
The pipeline is threefold, 1) point-to-mesh implicit reconstruction, 2) mesh-to-point rendering, and 3) zero-shot perception process.
First two stages are designed to simulate target domain's LiDAR configuration using source domains. For better granulity compared to using raw sparse LiDAR points, it utilizes implicit SDF representation.
The last stage is to use the simulated dataset for 3D detection method and perform zero-shot inference on the real dataset.

**Strengths:**

This paper suggests dataset generation/simulation pipeline that may enrich/adjust old annotated dataset to the new target domain.

**Weaknesses:**

1. Poor presentation
Overall, the placement of figures and tables is not aligned with the text, thereby the whole paper is difficult to follow.

2. Technical novelty & Writing
The paper mainly focuses on the dataset simulation process using old annotated dataset.
Therefore, the most of the methodology is restricted to step-by-step instructions, rather than providing theoretical insights or verifications.
I believe the paper's contribution on introducing new dataset simulation pipeline does not exceeds its lack of technical novelty.

3. Questionable dataset selection
Since Waymo dataset is more recent and contains more LiDAR sensors all around the vehicle, compared to nescenes or KITTI, wouldn't it be more plausible to simulate Waymo from KITTI, rather than KITTI from Waymo, to be more in coherence with the paper's motivation?
It seems like the pipeline only focuses on interchanging sensor configuration, using the richest point cloud information. Please elaborate.

On the minor note, I believe that this paper is more related to computer vision or robotics field than machine learning.

**Questions:**

Addressed in weakness section.

**Details Of Ethics Concerns:**

.

---

> ### Author Response · Authors · 2023-11-21
> **Response to Reviewer bhEL (Part 1/2)**
>
> We sincerely appreciate the reviewer's valuable comments. We have also tried our best to improve the quality and presentation of this manuscript, according to this reviewer’s comments.
>
> **Q1**: Poor presentation.
>
> **A1**: Thanks for your suggestions. The usage of English language and grammar of the revised manuscript has been double-checked thoroughly, and all the mistakes and errors which we can find have been revised and corrected. And the layout and order of the various tables and diagrams in the manuscripts have been better organized.
>
> **Q2**: Technical novelty: This paper focuses on the dataset simulation process.
>
> **A2**: In order to provide a clearer view of why and how the proposed ReSimAD is novel in autonomous driving community, we give an in-depth discussion of **the following three aspects**:
> - Motivation: First, based on the fact that the model pre-trained on source-domain still faces serious performance drop when deployed on the target domain (e.g., different weather or different LiDAR beam), this paper provides a new view of the source-domain reconstruction followed by the new-domain simulation. Basically, we assume that the data from source-domain are well-annotated and massive. However, direct training on the data from the source domain may result in the model overfitting to the data distribution of the source domain. Thus, we produce 3D mesh from the source domain to **decouple the domain characteristics**, and simulate the target-domain-like data according to the produced 3D mesh.  We are the first to achieve zero-shot target-domain 3D object detection.
> - Framework: Second, directly generating the 3D mesh using the off-the-shelf reconstruction method such as VDBFusion is challenging and cannot bring satisfactory performance gains for zero-shot 3D object detection, as demonstrated in Table 2 and Figure 6 of the revised manuscript. This is mainly because we assume that only point clouds are available and the 3D scene-level reconstruction based on 3D points is very challenging, due to: a) Sparse 3D points, and b) Insufficient reconstruction evaluation.
>
>   a) To address the issue of sparse 3D points, we aggregate all frames from the same scene sequence in Waymo dataset. Besides, we merge point clouds from side LiDARs to fill the blind spots around the vehicle, utilizing denser point clouds as input for reconstruction.
>
>   b) To address the issue of insufficient reconstruction evaluation, we render the Waymo points based on the generated 3D mesh, and calculate the Root Mean Square Error and Chamfer Distance between the simulated Waymo points and the original Waymo points. The results in Table 6 of the Appendix indicate that the sequences we used have low errors and exhibit high reconstruction quality.
> - Task: Finally, aiming to reduce the annotation cost of autonomous driving manufacturers for an unseen domain, we present the zero-shot target-domain 3D object detection task, where it is assumed that all annotated data from the source domain are available and no target domain data can be accessed. The presented task setting is practical, and as illustrated in Tables 2 and 4, we have verified that the proposed ReSimAD can achieve a satisfactory detection performance on the target domain compared to baseline models and UDA models.

---

> ### Author Response · Authors · 2023-11-21
> **Response to Reviewer bhEL (Part 2/2)**
>
> **Q3**: Questionable cross-domain setting selection.
>
> **A3**: Thanks for your concerns. However, we would like to clarify that the proposed cross-domain setting (namely, adaptation from Waymo to nuScenes) ****is rational****, due to the following reasons:
>
> 1) Fair comparison with the existing Unsupervised Domain Adaptation (UDA) methods: The existing UDA works [1, 2, 3, 4, 5] employ the Waymo-to-KITTI, Waymo-to-nuScenes setting to study the model cross-domain ability under the 3D scenario. In order to make fair comparisons with previous cross-dataset works, we align their cross-dataset setting, selecting the Waymo-to-KITTI, Waymo-to-nuScenes, and Waymo-to-ONCE as the baseline setting.
>
> 2) Reconstruction process: In order to achieve high-quality reconstruction results, we have to merge multiple frames sampled from the same 3D sequence to perform the reconstruction process, which is also inspired by UniSim and StreetSurf. Different from these works, we also utilize the side LiDARs to fill the blind spots around the vehicle to boost the reconstruction quality. Besides, we are the first to explore the so-called one-for-all setting, meaning that the source-domain reconstruction process is only performed once using the proposed ReSimAD, and simulated on different downstream datasets including KITTI, nuScenes, and ONCE.
>
> 3) ONCE is a challenging dataset, publicly available in 2022. We also select ONCE as the target domain, evaluating the cross-dataset performance of the proposed ReSimAD. On Waymo-to-ONCE setting, we also observe some interesting findings as reported in Table 2 of the manuscript: a) The existing UDA work, such as ST3D[1], only achieve 68.13\% mAP compared to 68.82\%  mAP of the source only (baseline result). This is mainly due to that the data distribution of ONCE dataset is very different from the Waymo dataset, including category definition and scene layout. b) Our ReSimAD exceeds the cross-dataset performance achieved by UDA work under the challenging Waymo-to-ONCE setting. Based on this, we believe that ReSimAD would pave the way for future exploration in zero-shot cross-dataset 3D study.
>
> According to the reviewer’s comments, we have supplemented detailed descriptions and explanations on Pages 7and 8 of the revised manuscript.
>
> **References**:
>
> [1] Jihan Yang, Shaoshuai Shi, Zhe Wang, Hongsheng Li, and Xiaojuan Qi. St3d: Self-training for unsupervised domain adaptation on 3d object detection. In Proceedings of the IEEE/CVF Conference on Computer Vision and Pattern Recognition, pp. 10368–10378, 2021.
>
> [2] Jiakang Yuan, Bo Zhang, Xiangchao Yan, Tao Chen, Botian Shi, Yikang Li, and Yu Qiao. Bi3d: Bi-domain active learning for cross-domain 3d object detection. In Proceedings of the IEEE/CVF Conference on Computer Vision and Pattern Recognition, pp. 15599–15608, 2023b.
>
> [3] Zhuoxiao Chen, Yadan Luo, Zheng Wang, Mahsa Baktashmotlagh, Zi Huang. Revisiting Domain-Adaptive 3D Object Detection by Reliable, Diverse and Class-balanced Pseudo-Labeling. In *Proceedings of the IEEE/CVF International Conference on Computer Vision* (pp. 3714-3726).
>
> [4] Yi Wei, Zibu Wei, Yongming Rao, Jiaxin Li, Jie Zhou, Jiwen Lu. Lidar distillation: Bridging the beam-induced domain gap for 3d object detection. In *European Conference on Computer Vision* (pp. 179-195).
>
> [5] Jiageng Mao, Shaoshuai Shi, Xiaogang Wang, Hongsheng Li. 3d object detection for autonomous driving: A review and new outlooks. arXiv preprint arXiv:2206.09474.

---

> > ### Comment · Reviewer_bhEL · 2023-11-23
> >
> > I thank the authors for their response.
> > I still think the main contribution of this paper is majorly focused on data simulation pipeline, which, still seems unfair since as the authors denoted, there exists sparsity issue for older LiDAR datasets.
> > However, I admit that other approaches and suggested benchmark follow same domain-shift scenarios, and is not the authors' fault.
> > I change my rating to 5. I am not against the acceptance of this paper, but I still believe there exists a lot of room for improvement.

---

> > > ### Author Response · Authors · 2023-11-23
> > > **Thanks for your rapid response**
> > >
> > > Thanks for your rapid response.
> > >
> > > - Firstly, for your concern of sparsity for older LiDAR dataset, we would like to clarify that, such zero-shot performance requires the underlying similarity of driving scenarios and a large amount of geometric observations (to form background 3d meshes).
> > > In selecting the dataset for source domain reconstruction, our primary focus is on whether the LiDAR has **a sufficiently large vertical Field of View (vFOV)**, at least equal to or larger than that of the target domain. If the source domain cannot capture information from a broader area, it will be incapable of acquiring reconstruction information related to the target domain's range.
> > >
> > >     &ensp;
> > >
> > >     This limitation manifests in the reconstruction process as artifacts or voids appearing in the grid in areas beyond the point cloud FOV of the source domain or in distant regions. To avoid incorporating these parts of the region into the target domain data during the LiDAR rendering process, we ensure that the target domain's vFOV is less than or equal to that of the source domain's LiDAR. The parameters of the LiDAR systems in Waymo, NuScenes, and Kitti are as follows[1, 2, 3].
> > >
> > >    &ensp;
> > >
> > >    |  Dataset |  | vFOV | |
> > >    | :---: | :--: | :----: | :----: |
> > >    | **Waymo** | *top* | [-17.6, 2.4] | |
> > >    |   |  *side**x4*** | [-90, 30] | |
> > >    | **KITTI** |  | [-24.9, 2]   | |
> > >    | **nuScenes** |   | [-30.67, 10.67] | |
> > >
> > >    As can be seen from the table, the Waymo dataset has four side LiDARs, which can greatly assist in the reconstruction of the surrounding roads and buildings on both sides of the vehicle. For the KITTI dataset, we consider that its vFOV coverage is relatively small, making it difficult to perform our domain transfer method. For the nuScenes dataset, we believe it can be adapted to our method, allowing for domain transfer to Waymo top LiDAR and KITTI, but not to the Waymo side LiDARs.
> > >
> > > &ensp;
> > >
> > > - Besides, we are **doing our best** to revise our manuscript according to your valuable suggestions, including: 1) the placement of figures and tables, 2) novelty clarification, and 3) dataset selection of employing Waymo to perform the scene-level reconstruction.
> > >
> > > &ensp;
> > >
> > > We believe that our ReconstructionSimulation-Perception pipeline can greatly help the industry reduce the cost of cross-domain deployment.
> > >
> > > &ensp;
> > >
> > > Best regards,
> > >
> > > Authors of Paper 1573
> > >
> > >
> > > [1] Pei Sun, Henrik Kretzschmar, Xerxes Dotiwalla, Aurelien Chouard, Vijaysai Patnaik, Paul Tsui, James Guo, Yin Zhou, Yuning Chai, Benjamin Caine, et al. Scalability in perception for autonomous driving: Waymo open dataset. In Proceedings of the IEEE/CVF conference on computer vision and pattern recognition, pp. 2446–2454, 2020.
> > >
> > > [2] Andreas Geiger, Philip Lenz, and Raquel Urtasun. Are we ready for autonomous driving? the kitti vision benchmark suite. In 2012 IEEE conference on computer vision and pattern recognition, pp. 3354–3361. IEEE, 2012.
> > >
> > > [3] Holger Caesar, Varun Bankiti, Alex H Lang, Sourabh Vora, Venice Erin Liong, Qiang Xu, Anush Krishnan, Yu Pan, Giancarlo Baldan, and Oscar Beijbom. nuscenes: A multimodal dataset for autonomous driving. In Proceedings of the IEEE/CVF conference on computer vision and pattern recognition, pp. 11621–11631, 2020.

---

> > > ### Author Response · Authors · 2023-11-23
> > > **Paper Update for addressing your concerns**
> > >
> > > To further address the concerns from the reviewer towards: 1) paper presentation, 2) dataset selection, and 3) core contribution, we have taken the following actions and have uploaded a new version of the manuscript.
> > >
> > > - **For the paper presentation**, the revised manuscript has been double-checked thoroughly. For example, the layout and order of tables and figures (including Tables 1 and 3, Figures 4 and 5) in the manuscripts have been better reorganized, in order to ensure that the tables or figures are aligned with the corresponding text. Besides, all the mistakes and errors which we can find have been revised and corrected.
> > >
> > > - **In order to avoid ambiguity**, we have supplemented a subsection on Page 6 of the revised manuscript to further explain why we use Waymo as the source domain and the reason for employing such cross-dataset settings.
> > >
> > > - **For the core contribution**, we agree with the reviewer's comment that, data simulation pipeline is our main contribution. But we would like to clarify that in autonomous driving community, implementing such a data simulation process is challenging and unprecedented, which is mainly due to: a) sparse 3D points, and b) insufficient reconstruction evaluation. As stated in [Response to Reviewer bhEL (Part 1/2)
> > > ](https://openreview.net/forum?id=1d2cLKeNgY&noteId=Oiiboe4qBr), we tackled such challenges and are the first to boost the zero-shot detection accuracy for the target domain. This is practical for industrial people, as commented from Reviewer kWr7.
> > >
> > > &ensp;
> > >
> > > At last but not least, we would like to take this opportunity to thank the reviewer's insightful comments, which greatly helped us to improve the technical quality and the presentation of this manuscript. It is appreciated that you could consider these modifications and rapid actions of our manuscript.
> > >
> > > &ensp;
> > >
> > > Sincerely Yours,
> > >
> > > Authors of Paper 1573.

---

### Official Review · Reviewer_kWr7 · 2023-10-31

**Soundness:** 2 fair
**Presentation:** 1 poor
**Contribution:** 2 fair
**Rating:** 6
**Confidence:** 3

**Summary:**

This is a system paper. The authors study a weakened domain generalization setting for 3d object detection from lidar point clouds. For example, training only on Waymo dataset and only accessing the target domain's lidar statistics, the setting pursues good performance on nuScenes. Since information about the target domain is not completely unknown, this is a weakened domain generalization setting. The proposal is to reconstruct the mesh using aggregated LIDAR (or RGB? as NeuS is mentioned). Then the background mesh is put into Carla and car assets are placed according to object size matching. The lidar signal is simulated using the composed scene in Carla. Then the authors train detectors using these simulated data and show the results out-perform the UDA baseline.

**Strengths:**

+ The idea generally makes sense and the authors benchmark it in a large-scale, showing meaningful margins.

**Weaknesses:**

- The biggest issue is a lack of clarify, for both the mesh reconstruction and the sampling part:
* The reconstruction does not define input, output and losses formally. The only equation is depth rendering. Aggregated lidar piont clouds are used for training? (this is highlighted in bold texts) The authors mention NeuS and I am not sure whether RGB rendering losses are used.
* Lidar rendering is difficult and there are some sphiscated methods to simulate second-returns [A]. Again the lidar rendering part does not contain any formal mathematical exposition so I cannot understand what the lidar rendering formulation is.
Since two major algorithmic parts are not understandable, I am rating presentation as poor.

- Let alone the presentation issue about algorithms, I will just assume that these two parts invoke some black-box functions and only consider the system. In this regard, I find the mesh of poor quality. This is understandable as recent papers from my group can only reconstruct meshes from lidar with similar quality. Having that said, I cannot understand why points generated from them are meaningful for detection. The authors should present a systematic evalution for mesh quality (Table.6 does not make sense to me) and rendered point cloud quality. Some comparisons are also needed, e.g., comparing with point clouds rendered from VDBF?
This concern makes me rate soundness as fair.

Minor but still confusing issues:
- Fig.2 is confusing. I cannot understand what the differences are except for the figure color.
- Table.1 gives literally no additional information since target domains are already mentioned in texts. And Waymo is also target domain？

[A] Neural LiDAR Fields for Novel View Synthesis

**Questions:**

See above.

---

> ### Author Response · Authors · 2023-11-21
> **Response to Reviewer kWr7**
>
> Thank you for your comments. We provide discussions and explanations about your concerns as follows.
>
> **Q1**: Clarifying issues related to mesh reconstruction.
>
> **A1**: The main contribution claimed in this paper is the unified reconstruction-simulation-perception paradigm that addresses the domain shift issue. Considering previous work, StreetSurf [1], current advanced reconstruction methods are already capable of achieving high-quality point cloud rendering in the target domain, and the specific reconstruction methods are not our main contribution. In the mesh reconstruction section, we opt for a LiDAR-only setting, which allows us to be unaffected by lighting conditions and to obtain a richer and larger dataset of the scene. Drawing inspiration from StreetSurf [1], **the reconstruction input is derived from LiDAR rays, and the output is the predicted depth**.
>
> On each sampled LiDAR beam $\mathbf{r}\_{\text{lidar}}$, we apply a logarithm L1 loss on $\hat{D}^{(\mathrm{cr}, \mathrm{dv})}$, the rendered depth of the combined close-range and distant-view model:
> $$
> {\mathscr{L}}\_{geometry} = \ln \\left ( |\hat{D}^{(\mathrm{cr}, \mathrm{dv})} (\mathbf{r}\_{lidar}) - D(\mathbf{r}\_{lidar }) | + 1\\right )
> $$
>
> To achieve improved reconstruction quality, we aggregate all frames from the same scene sequence in the Waymo dataset. Additionally, we merge point clouds from side LiDARs to fill the blind spots around the vehicle, utilizing denser point clouds as input for reconstruction.
>
> Simultaneously, we have conducted a quantitative evaluation of the reconstruction. Within the target domain LiDAR FOV (which is smaller than the source domain LiDAR FOV), we sample and calculate the Root Mean Square Error and Chamfer Distance for reconstruction errors. The results in Table 6 of the ****Appendix**** indicate that, the sequences we used have low errors and exhibit high reconstruction quality.
>
> Due to space constraints, the evaluation of reconstruction quality and the explanation of LiDAR rendering are placed in the Appendix section. We greatly appreciate your review of this material.
>
> &ensp;
>
> **Q2**: Clarifying issues related to LiDAR rendering. What the lidar rendering formulation is?
>
> **A2**: In the LiDAR rendering section, we utilize the **official specifications** of the LiDAR used in the target domain to complete the modeling of scanning characteristics, while also considering the simulation of physical properties. The key technical details of this part are fully elaborated in previous work on LiDAR simulation [2]. A brief overview is also provided in Appendix A.2.
>
> &ensp;
>
> **Q3**: Comparation with point clouds rendered from VDBF.
>
> **A3**: Regarding your mention of comparing mesh quality with the VDBF method, it is true that we have only conducted a preliminary comparison in terms of visualization. However, considering the numerous voids and artifacts introduced by VDBF, it is evident that this would impact the quality of LiDAR rendering in our downstream section, making it challenging to achieve the results of our current reconstruction method.
>
> &ensp;
>
> **Q4**: Why points generated from mesh are meaningful for detection?
>
> **A4**: We decouple the sensor characteristics of the source domain through mesh reconstruction and employ sensors conforming to the target domain characteristics during the LiDAR rendering. This leads to generated points with smaller domain differences from the target domain, which is more conducive to learning the features of the target domain point cloud. Consequently, this can greatly enhance the performance of downstream tasks.
>
> &ensp;
>
> **Q5**: Confusion issue of Figure 2.
>
> **A5**: With regards to Figure 2, our original intention was to intuitively illustrate that, the LiDAR data obtained through our method closely resembles the real data, without any noticeable variations between domains. We acknowledge your suggestion and will remove this figure in the upcoming updated version.
>
> &ensp;
>
> **Q6**: Confusion issue of Table 1.
>
> **A6**: As for Table 1, which discusses the Waymo-like dataset, it refers to the target domain Waymo dataset obtained by rendering on the mesh reconstructed from the source domain Waymo dataset. We use this to **assess reconstruction errors**, as detailed in Appendix Table 6. If this causes any ambiguity, we will provide a more detailed explanation in the legend of the upcoming version.
>
> **References**:
>
> [1] Jianfei Guo, Nianchen Deng, Xinyang Li, Yeqi Bai, Botian Shi, Chiyu Wang, Chenjing Ding, Dongliang Wang, and Yikang Li. Streetsurf: Extending multi-view implicit surface reconstruction to street views. arXiv preprint arXiv:2306.04988, 2023.
>
> [2] Xinyu Cai, Wentao Jiang, Runsheng Xu, Wenquan Zhao, Jiaqi Ma, Si Liu, and Yikang Li. Analyzing infrastructure lidar placement with realistic lidar simulation library. In 2023 IEEE International Conference on Robotics and Automation (ICRA), pp. 5581–5587, 2023. doi: 10.1109/ICRA48891.2023.10161027.

---

> ### Comment · Reviewer_kWr7 · 2023-11-22
> **Quick Feedback.**
>
> Q1. Thanks for clarification. Now I understand the part is based upon another paper [1].
>
> Q2. Thanks for clarification. Now I understand the part is based upon another paper [2].
>
> Q3. Thanks for clarification. Now I understand that the VDBF quality is so bad that it does not deverse a quantitative comparisons.
>
> Q4. Now I understand that although the mesh quality of ReSimAD seems low, e.g., bottom row of Fig.5, it may be enough for LiDAR simulation.
>
> Q5&6. Thanks, although I don't see the updated version. There are still 24 hours. Looking forward to these updates before the update panel closes.

---

> ### Author Response · Authors · 2023-11-22
> **Thanks for your rapid response**
>
> Dear Reviewer:
>
> Thanks for your precious time and valuable suggestions to improve the quality of our paper. We have also carefully revised the manuscript, with modifications marked with blue text.
>
> We hope that our revision makes our presentation clearer and covers your concerns. We are happy to discuss any remaining questions about our work.
>
> Best regards,
> Authors of Paper 1573

---

> > ### Comment · Reviewer_kWr7 · 2023-11-22
> > **feedback**
> >
> > I have raised the recommendation to score to ba. This system paper may have some impact on industrial people.

---

### Official Review · Reviewer_7ZCS · 2023-11-10

**Soundness:** 3 good
**Presentation:** 3 good
**Contribution:** 3 good
**Rating:** 6
**Confidence:** 3

**Summary:**

This paper proposes RESIMAD, which aims to base source 3D reconstruction for target-domain point clouds data generation. Specifically, using implicit neural fields with accumulated point clouds per sequence, a high-quality source-domain 3D mesh could be extracted as background of AD scenes. Foreground synthetic assets like vehicles are added within simulation guided by source GT information. Therefore, target-domain point clouds could be further extracted directly or rectified with domain-specific sensor specs. Experiments compared against UDA baseline and pre-training effectiveness demonstrate the potential usage of RESIMAD.

**Strengths:**

- The overall paper is well motivated and easy to follow. The RESIMAD pipeline of is composed of three key components and clearly discussed with adequate implementation description.

- The experiments on several Waymo to other datasets setup shows the effectiveness of RESIMAD.

**Weaknesses:**

1. Effectiveness on larger detection models. One potential benefit of a ‘reconstruction, simulation and perception’ pipeline is that numerous data could be generated/rendered. As only a typical point RCNN baseline model is chosen for evaluation, it would further highlight the advantages of such methods considering more powerful and data-hungry models (e.g., ViT, DETR like detection models).

2. Only ‘zero-shot’ performance is given. It would still be valuable to see whether RESIMAD could benefit from increasingly more target-domain information, from sensor specs (almost zero shot), to few-shot set-up or even with more target samples available)

Above tow points also applies to the pre-training experiment in Table3.

3. Comparison against more recent DA/UDA methods. UDA is a popular topic attracting much attentions, it is expected to compare against more recent SOTA DA/UDA approaches to further support the evaluation.

4. Although an almost ‘zero-shot’/unsupervised DA could be achieved, such zero-shot performance, I would say, requires the underlying similarity of driving scenarios and a large amount of geometric observations (to form background 3d meshes). I am wondering if the source domain shifts from Waymo to nuScenes or KITTI, will the proposed method still able to work well?

5. Related to last point, one of my main concern is that the acquisition of 3D meshes. It may require a lot of computational costs and multi-lidar sensor specs for pre-training (per-scene optimization of LINR) to get relatively complete and accurate 3d meshs?

Effectiveness and contributions of the proposed methods could be further improved in the above aspects in my opinion. I would like to hear from from authors in the rebuttal phase.

**Questions:**

Please see the weaknesses section above.

---

> ### Author Response · Authors · 2023-11-21
> **Response to Reviewer 7ZCS (Part 1/2)**
>
> **Q1**: Effectiveness on larger detection models.
>
> **A1**: Thanks for your valuable comment. Considering that we are the first work to explore the zero-shot cross-dataset ability of 3D detection model, we align with previous cross-dataset works [1, 2, 3, 4], where they often employ PV-RCNN and PV-RCNN++ as the baseline models (as reported in Table 1 of their papers). The purpose of employing the PV-RCNN model is to make a fair comparison with previous cross-dataset works [1, 2, 3, 4].
>
> **References**:
>
> [1] Jihan Yang, Shaoshuai Shi, Zhe Wang, Hongsheng Li, and Xiaojuan Qi. St3d: Self-training for unsupervised domain adaptation on 3d object detection. In Proceedings of the IEEE/CVF Conference on Computer Vision and Pattern Recognition, pp. 10368–10378, 2021.
>
> [2] Jiakang Yuan, Bo Zhang, Xiangchao Yan, Tao Chen, Botian Shi, Yikang Li, and Yu Qiao. Bi3d: Bi-domain active learning for cross-domain 3d object detection. In Proceedings of the IEEE/CVF Conference on Computer Vision and Pattern Recognition, pp. 15599–15608, 2023b.
>
> [3] Zhuoxiao Chen, Yadan Luo, Zheng Wang, Mahsa Baktashmotlagh, Zi Huang. Revisiting Domain-Adaptive 3D Object Detection by Reliable, Diverse and Class-balanced Pseudo-Labeling. In *Proceedings of the IEEE/CVF International Conference on Computer Vision* (pp. 3714-3726).
>
> [4] Yi Wei, Zibu Wei, Yongming Rao, Jiaxin Li, Jie Zhou, Jiwen Lu. Lidar distillation: Bridging the beam-induced domain gap for 3d object detection. In *European Conference on Computer Vision* (pp. 179-195).
>
> **Q2**: It would still be valuable to see whether the proposed method could benefit from increasingly more target-domain information.
>
> **A2**: We are very grateful to the reviewer for the insightful suggestion. According to the reviewer’s comment, we have illustrated the experimental results using ReSimAD under the fully-supervised target-domain setting. As reported in Table 3 of the revised manuscript, ReSimAD can provide better network initialization parameters, compared to the baseline trained from scratch. Specifically, PV-RCNN model is firstly trained from scratch on the target domain (nuScenes) using fully annotated data, and its performance is 53.07\% on the target domain. By comparison, PV-RCNN model is initialized using the checkpoint pre-trained by the proposed ReSimAD, and fine-tuned on the target domain. The target-domain performance ($AP_{BEV}$) can outperform the one trained from scratch, achieving ****54.48\%****.
>
> **Q3**: Expected to compare against more recent SOTA DA/UDA approaches to further support the evaluation.
>
> **A3**: Thanks for the valuable recommendation. As you observe in Table 1, there is still a performance gap between our ReSimAD (81.01\%) and the UDA method (84.10\%) on the target domain (KITTI). This is mainly due to that, our ReSimAD aims to zero-shot cross-dataset setting, where it is assumed that ****no**** data from the target domain are **available**. But for the UDA method, they assume that the data from the target domain are available for model training. Considering this, we select the ST3D [1], the representative 3D UDA method, to show the cross-dataset detection accuracy of the network under different input conditions (zero-shot or UDA settings). Besides, we select two zero-shot methods, which are CARLA-default and Sensor-like, as the comparison baselines. The experimental results are shown in Table 2.
>
> **Reference**:
>
> [1] Jihan Yang, Shaoshuai Shi, Zhe Wang, Hongsheng Li, and Xiaojuan Qi. St3d: Self-training for unsupervised domain adaptation on 3d object detection. In Proceedings of the IEEE/CVF Conference on Computer Vision and Pattern Recognition, pp. 10368–10378, 2021.

---

> > ### Author Response · Authors · 2023-11-21
> > **Response to Reviewer 7ZCS (Part 2/2)**
> >
> > **Q4**: Performance of the proposed method when the source domain changes to nuScenes or KITTI Domain.
> >
> > **A4**: Thanks for your insightful finding. We totally agree with the reviewer that, such zero-shot performance requires the underlying similarity of driving scenarios and a large amount of geometric observations (to form background 3d meshes). Based on such an assumption, ReSimAD (source-domain reconstruction and target-domain simulation) can achieve a satisfactory zero-shot detection accuracy on the target domain.
> >
> > Besides, in selecting the dataset for source domain reconstruction, our primary focus is on whether the LiDAR has a sufficiently large vertical Field of View (vFOV), at least equal to or larger than that of the target domain. If the source domain cannot capture information from a broader area, it will be incapable of acquiring reconstruction information related to the target domain's range. This limitation manifests in the reconstruction process as artifacts or voids appearing in the grid in areas beyond the point cloud FOV of the source domain or in distant regions. To avoid incorporating these parts of the region into the target domain data during the LiDAR rendering process, we ensure that the target domain's vFOV is less than or equal to that of the source domain's LiDAR.
> >
> > The parameters of the LiDAR systems in Waymo, NuScenes, and Kitti are as follows[1, 2, 3].
> >
> > |  Dataset |  | vFOV | |
> > | :---: | :--: | :----: | :----: |
> > | **Waymo** | *top* | [-17.6, 2.4] | |
> > |   |  *side**x4*** | [-90, 30] | |
> > | **Kitti** |  | [-24.9, 2]   | |
> > | **nuScenes** |   | [-30.67, 10.67] | |
> >
> > As can be seen from the table, the Waymo dataset has four side LiDARs, which can greatly assist in the reconstruction of the surrounding roads and buildings on both sides of the vehicle. For the Kitti dataset, we consider that its vFOV coverage is relatively small, making it difficult to perform our domain transfer method. For the nuScenes dataset, we believe it can be adapted to our method, allowing for domain transfer to Waymo top LiDAR and Kitti, but not to the Waymo side LiDARs.
> >
> > **References**:
> >
> > [1] Pei Sun, Henrik Kretzschmar, Xerxes Dotiwalla, Aurelien Chouard, Vijaysai Patnaik, Paul Tsui, James Guo, Yin Zhou, Yuning Chai, Benjamin Caine, et al. Scalability in perception for autonomous driving: Waymo open dataset. In Proceedings of the IEEE/CVF conference on computer vision and pattern recognition, pp. 2446–2454, 2020.
> >
> > [2] Andreas Geiger, Philip Lenz, and Raquel Urtasun. Are we ready for autonomous driving? the kitti vision benchmark suite. In 2012 IEEE conference on computer vision and pattern recognition, pp. 3354–3361. IEEE, 2012.
> >
> > [3] Holger Caesar, Varun Bankiti, Alex H Lang, Sourabh Vora, Venice Erin Liong, Qiang Xu, Anush Krishnan, Yu Pan, Giancarlo Baldan, and Oscar Beijbom. nuscenes: A multimodal dataset for autonomous driving. In Proceedings of the IEEE/CVF conference on computer vision and pattern recognition, pp. 11621–11631, 2020.
> >
> > **Q5**: The computational and sensor requirements for mesh reconstruction.
> >
> > **A5**: We adopt the reconstruction method from StreetSurf [1] to complete the mesh reconstruction section. For each scene, we train LINR from scratch without any pre-training. Under the LiDAR only setting, the computational cost required for each sequence's reconstruction task is:
> >
> > - <=2 hrs training time on a single RTX3090
> > - ~16 GiB GPU Memory
> > - \>24 GiB CPU Memory (caching data to speed up)
> >
> > Regarding the sensor specification you mentioned, it is not necessary to provide the sensor spec during the reconstruction section. However, in the LiDAR rendering section, the sensor spec for the corresponding target domain is required. This aspect has already been included in the Simulation Library proposed in [2].
> >
> > **References**:
> >
> > [1] Jianfei Guo, Nianchen Deng, Xinyang Li, Yeqi Bai, Botian Shi, Chiyu Wang, Chenjing Ding, Dongliang Wang, and Yikang Li. Streetsurf: Extending multi-view implicit surface reconstruction to street views. arXiv preprint arXiv:2306.04988, 2023.
> >
> > [2] Xinyu Cai, Wentao Jiang, Runsheng Xu, Wenquan Zhao, Jiaqi Ma, Si Liu, and Yikang Li. Analyzing infrastructure lidar placement with realistic lidar simulation library. In 2023 IEEE International Conference on Robotics and Automation (ICRA), pp. 5581–5587, 2023. doi: 10.1109/ICRA48891.2023.10161027.

---

> > > ### Comment · Reviewer_7ZCS · 2023-11-23
> > > **Response to authors' feedback**
> > >
> > > First I appreciate the detailed feedback from authors.
> > > As mentioned by other reviewers, a lot of implementation and components details are not mentioned and cause confusion when understanding the overall pipeline of ReSimAD, e.g., mesh reconstruction/preparation. I hope necessary discussions and analysis could be added into a revised version to strengthen the writing quality and contributions.
> > >
> > > Secondly, after this round of discussion, I better understand the motivations and limitations of the proposed method regarding its applicable asepcts.
> > >
> > > Overall, the rebuttal addressed most of my concerns about detailed information and hope all helpful materials could be added into revision. I would like to increase my score to BA considering such both clarification and inherent limitations of the proposed method.

---

> > > > ### Author Response · Authors · 2023-11-23
> > > > **Thanks for your approval and rapid response**
> > > >
> > > > Thanks for your approval and response.
> > > >
> > > >
> > > > According to your suggestion, the clarification and inherent limitations of the proposed ReSimAD will be added to the revised manuscript to avoid confusion. All adjustments will be updated in the revised manuscript before the deadline.
> > > >
> > > >
> > > > Best regards,
> > > >
> > > > Authors of Paper 1573

---

### Author Response · Authors · 2023-11-23
**General Response**

Dear AC and Reviewers,

Many thanks for your valuable comments and insightful suggestions to improve the quality of our original manuscript. We have provided detailed discussions and updated our manuscript to cover reviewers’ concerns. Here is a summary of what we have done in the rebuttal phase.

- We enhance the readability of the revised manuscript. For example, Figures 1 and 3 have been redrawn to clearly illustrate the pipeline of the proposed ReSimAD. Besides, the formatting of this paper (including Tables 1, 2, and 3, Figures 4 and 5) has been carefully adjusted. Besides, all the mistakes and errors which we can find have been revised and corrected.

- We supplement more technical descriptions including: a) the definition of input and output for point-to-mesh reconstruction, b) the computational requirement for mesh reconstruction, c) point cloud registration for consolidating all frames, and d) traffic flow reconstruction.
- We provide more discussions on our motivation and novelty, where we are the first to achieve a unified reconstruction-simulation-perception paradigm, which is challenging but practical for industrial applications.
- We provide necessary elaborations and supporting evidence. For example, explanations of point cloud registration when consolidating all frames from the corresponding sequence have been included in the revised manuscript. Besides, the details regarding the sample quantity and other experimental settings have been supplemented on Appendix.
- We make further clarification on the cross-dataset experimental setting used in our manuscript. For example, we have supplemented a subsection on Page 6 of the revised manuscript to further explain why we use Waymo as the source domain and the reason for employing such cross-dataset settings.

&ensp;

At last but not least, we would like to take this opportunity to thank the four anonymous reviewers again for their insightful comments and valuable suggestions, which greatly helped us to improve the technical quality and the presentation of this manuscript. Moreover, we thank the Area Chair, again, for reading the responses.

&ensp;

Sincerely Yours,

Authors of Paper 1573.

---

### Meta-Review · Area_Chair_MTTh · 2023-12-06

**Metareview:**

This paper a new pipeline for 3D domain shift in the context of autonomous driving.  The pipeline is threefold: point-to-mesh implicit reconstruction, mesh-to-point rendering, and zero-shot perception. The first two modules are designed to simulate target domain’s LiDAR configurations using source domains while the last module uses the simulated dataset for 3D detection and zero-shot inference on the real dataset.  The concern unanimously raised by the reviewers was poor writing.  The contribution of this work and technical details of the proposed method were not clear. Through the rebuttal and discussion between the authors and the reviewers, this concern was resolved and the manuscript was revised, accordingly.  The readability of the paper was enhanced.  Although the domain-shift scenarios, namely, Waymo-to-KITTI/nuScenes, may also be evaluated in the reverse way, this work will have some impact on industrial people.  Therefore, this paper is deserved to be accepted.

**Justification For Why Not Higher Score:**

Although this paper proposes a new framework for 3D domain shift, this is something like a system-oriented paper.  Novelty is not enough to be oral or spotlight.

**Justification For Why Not Lower Score:**

Although this paper proposes a new framework for 3D domain shift, this is something like a system-oriented paper.  Novelty is not enough to be oral or spotlight.

---

### Decision · Program_Chairs · 2024-01-16

Accept (poster)